# Oxidative Damage of Mussels Living in Seawater Enriched with Trace Metals, from the Viewpoint of Proteins Expression and Modification

**DOI:** 10.3390/toxics8040089

**Published:** 2020-10-18

**Authors:** Georgia G. Kournoutou, Panagiota C. Giannopoulou, Eleni Sazakli, Michalis Leotsinidis, Dimitrios L. Kalpaxis, George P. Dinos

**Affiliations:** 1Department of Biochemistry, School of Medicine, University of Patras, 26504 Patras, Greece; p.giannopoulou@upnet.gr (P.C.G.); dim.kalpax@gmail.com (D.L.K.); 2Laboratory of Public Health, School of Medicine, University of Patras, 26504 Patras, Greece; elsazak@upatras.gr (E.S.); micleon@upatras.gr (M.L.)

**Keywords:** copper, mercury, cadmium, oxidative stress, protein carbonylation, translation factors, oxidative stress biomarkers

## Abstract

The impact of metals bioaccumulation in marine organisms is a subject of intense investigation. This study was designed to determine the association between oxidative stress induced by seawater enriched with trace metals and protein synthesis using as a model the mussels *Mytilus galloprovincialis*. Mussels were exposed to 40 μg/L Cu, 30 μg/L Hg, or 100 μg/L Cd for 5 and 15 days, and the pollution effect was evaluated by measuring established oxidative biomarkers. The results showed damage on the protein synthesis machine integrity and specifically on translation factors and ribosomal proteins expression and modifications. The exposure of mussels to all metals caused oxidative damage that was milder in the cases of Cu and Hg and more pronounced for Cd. However, after prolonged exposure of mussels to Cd (15 days), the effects receded. These changes that perturb protein biosynthesis can serve as a great tool for elucidating the mechanisms of toxicity and could be integrated in biomonitoring programs.

## 1. Introduction

Bivalves have been widely used as bioindicators in monitoring aquatic pollution, with mussels attracting most of the interest. Mussels accumulate trace metals and other organic and inorganic pollutants from the seawater via filtration, thereby providing an integrative measure of the concentration and bioavailability of seawater pollutants [1,2]. The survival of mussels in the aquatic environment depends on their ability to sense and respond to biotic and/or abiotic changes, such as exposure to metal insults. Such changes perturb cellular homeostasis and cause cellular damages. For instance, copper is a ubiquitous trace metal of vital importance, serving as a cofactor in many metalloenzymes. However, large concentrations of Cu become harmful [3,4] because of its propensity to mediate the formation of reactive oxygen species (ROS) [4,5]. In addition, Cu reacts with thiol groups; thus, it is able to bind cysteine and inactivates proteins [6], induces the expression of metallothionein (MT) genes [7,8], causes lysosomal membrane destabilization [9,10,11,12], and perturbs the nucleus integrity [10,11,12,13,14,15]. Similarly, Cd inactivates many functional proteins by inducing widespread misfolding and aggregation through binding to S, N, and O protein atoms [16], destabilizes lysosomal membranes [10], and induces the expression of MT genes [8]. Cd is additionally capable of reacting directly with free-SH groups, thus inhibiting the activity of catalase and glutathione (GSH) reductase, and either directly or indirectly depleting the cellular pools of GSH [3]. It can also inactivate essential proteins by displacing Zn or Ca from the active sites [4], induce apoptosis and necrosis [17], and facilitate DNA mutagenesis through mismatch repair [18]. Finally, Hg is particularly prone to reacting with free-SH groups, inhibiting thereby the function of various proteins rich in thiols and depleting GSH levels [19]. Among trace metals, Hg is the most genotoxic agent, following the genotoxic potential order: Hg > Cu > Cd [13].

Apart from the above toxic effects, Cu, Hg, and Cd are also characterized by their ability to disturb the cellular balance in redox systems. Cu and Hg can do this directly via Fenton or Haber–Weiss reaction [20], while Cd does it indirectly through glutathione depletion [21]. In the last few years, toxicological studies have started to investigate the functional and structural aberrations in the translation machinery upon the exposure to trace metals [22,23,24]. It has been realized that a general translation response to metal toxicity is the repression of global protein synthesis, inducing in parallel specific proteins overexpression, including mostly antioxidant defense and heat shock proteins [11,14,15,25,26,27]. The toxicity is also correlated with gene expression and regulation. Alterations in gene expression under oxidative stress have been extensively analyzed in mollusks through RNA profiling techniques, microarrays, and RNA sequencing [28,29,30,31,32].Moreover, translational regulation generally contributes to quick responses related to the maintenance of proteome homeostasis, in contrast to transcriptional regulation that is mostly associated with long-term changes in cell physiology [33,34,35]. It has been also recognized that the molecular mechanisms underlying the metal toxicity on translation are associated with the vulnerability of the ribosomal components to oxidative damage [10,36,37,38,39]. Proteins can scavenge up to 75% of oxidative insults, with ribosomal proteins being the most likely class of proteins to be oxidized [39,40]. Except for sulfoxide formation, which can be reversible under circumstances, most protein damage is non-repairable and can lead to numerous deleterious consequences in the cellular metabolism [41].

Here, we study the trace metal oxidative damage in mussels after their exposure in seawater enriched with each of the metal ions Cu, Hg, and Cd. We tried to reproduce the natural seawater environment polluted with high concentrations of trace metals. Mussels grew for defined periods and the stress was evaluated by measuring established biomarkers, comprising either antioxidant barriers, such as superoxide dismutase activity, metallothioneinlevels, glutathione, or oxidative damage indicators, such as micronucleus frequency, superoxide radical production, labilization period of the lysosomal membrane, lipid peroxidation, and carbonylated proteins. In parallel, crucial protein synthesis factors were determined as well as ribosomal proteins abundance and integrity, and their correlation with the oxidative stress biomarkers were examined. All biomarkers were measured in a fraction containing total cytosolic proteins from homogenized digestive glands and additionally in a subcellular ribosome wash fraction rich in translation factors and lastly in crude ribosomal proteins extracted from 80S ribosomes. According to our data, trace metals are serious oxidative stressors in aquatic ecosystems, and all established oxidative biomarkers can directly be correlated with protein synthesis machine integrity and translational reprogramming. The last can be used as a new biomonitoring tool concerning the trace metal pollution.

## 2. Materials and Methods

### 2.1. Exposure of Mussels to Trace Metals

Mussels *M. galloprovincialis* of narrow size (6.0 ± 0.5 cm), not at reproductive state, were provided by a marine farm (Poseidon Co., Mandros; Galaxidi, Greece), transported to the laboratory, and acclimated for 1 week, at constant temperature (18 °C) in tanks containing natural and non-polluted seawater, which were previously filtered and sterilized with UV light. After acclimation, mussels were exposed to 40 μg/L Cu, 30 μg/L Hg, or 100 μg/L Cd for 5 or 15 days, which were added as divalent chloride salts after every change of seawater, and put in tanks under continuous aeration and a natural photoperiod [14]. The concentrations of trace metals exposure were selected to be high enough to reach the appropriate level after a short time of treatment, and the incubation time was also kept short to maintain high concentrations without mortality [14]. Mussels were fed daily in two doses (38 mg of food per dose; PROCORAL, PHYTON Tropic Marin, Wartenberg, Germany). The seawater in tanks was exchanged every 48h, while the desired metal concentration was maintained. After the exposure period, 20 mussels were dissected, and several equally mixed pools of gills and digestive glands were immediately frozen and stored at −80 °C until use. Repetitions of the assays were performed using different pools to quantify biological replicates.

### 2.2. Biomarkers’ Analyses

#### 2.2.1. Metal Concentration in Mussels

Metals were determined in a composite sample of digestive glands excised from 20 mussel specimens. The digestive glands of mussels were freeze-dried and pulverized. Approximately a 0.5 g sample, accurately weighted, was digested with 7 mL of supra-pure 65% HNO_3_ and 2 mL of 30% H_2_O_2_ by using a microwave-assisted closed wet digestion (Ethos Touch). The obtained solutions were diluted to a final volume of 10 mL with MilliQ water. Atomic absorption spectrometry (AAS) was employed in graphite mode (GF-AAS) for the determination of Cd and Cu, while Hg was determined via cold vapor AAS (Shimadzu AA-6300 system, equipped with graphite furnace GFA-EX7i and Hydride Vapor Generator HVG-1). Calibration was performed via standard solutions subjected to the same digestion procedure. Detection limits were calculated to be 0.004 μg/g for Cd, 0.04 μg/g for Cu, and 0.006 μg/g for Hg. Precision was estimated at 5–7% by replicate measurements. The recovery of known trace element amounts added to the samples before wet digestion varied from 90% to 103%. The quality assurance of metal analyses was further checked by using one certified reference material, the IAEA-436 biota sample, provided by the IAEA’s Marine Environment Studies Laboratories (MESL). The determined values did not differ by more than 5% from the certified ones.

For the determination of Cd and Cu in seawater, a pre-treatment step was employed. The samples were subjected to extraction with ammonium pyrrolidine dithiocarbamate (APDC)/methyl isobutyl ketone (MIBK) and then back-extracted in 0.3 M nitric acid. For Hg, no pre-treatment step was required. Measurements by Atomic Absorption Spectrometry as described above, were followed.

#### 2.2.2. Oxidative Damage Indicators

Lysosomal membrane stability was calculated with a cytochemical technique by measuring the time in minutes required to destabilize the lysosomal membranes under acid conditions [42]. This protocol is a cytochemical procedure and is based on the evaluation of the activity of N-acetyl-β-hexosaminidase, which is a lysosomal enzyme. The Labilization Period (LP) was determined by microscopic assessment of the pre-treatment time required to produce maximal staining intensity in a series of digestive gland sections.

Micronucleus (MN) frequency was determined in gill cells, according to Bolognesi and Fenech using a mussel cytome assay protocol [43]. Briefly, aliquots of cellular pellet of mussel gills were fixed in methanol:acetic acid (3:1), spread on slides, and stained with 3% Giemsa. A total of 2000 cells with preserved cytoplasm were scored under oil immersion at 1000× magnification. The relative occurrence of micronuclei can provide an indication of accumulated genetic damage in marine organisms even during short phases of contamination.

Lipid peroxidation in digestive gland lysates was estimated by determining the thiobarbituric reactive substances (TBARS), which were based on the reaction with the chromogenic reagent, N-methyl-2-phenylindole, with malondialdehyde and 4-hydroxyalkenals [44]. TBARS were determined by fluorescence using λex 535 nm and λem 550 nm. Results were expressed in nanomoles of malondialdeyde (MDA) equivalents produced per mg of tissue protein. Proteins in the digestive gland lysates and/or fractions were measured by the Bradford’s method, as modified by Grintzalis et al. [45].

Superoxide radical (O_2_^−^) production in vivo was measured by reaction with dihydroethidine injected into the mussels and circulated internally for 30 min using an ultrasensitive fluorescent assay for the in vivo quantification of superoxide radicals according to Georgiou et al. [46] and were expressed in pmoles produced per mg tissue protein in 30 min.

#### 2.2.3. Anti-Oxidant Barrier

Metallothionein (MT) content in digestive gland lysates was measured following the protocol and assumptions made by Viarengo et al. [47]. The method is briefly based on a metallothionein fractionation and SH residue content quantification by a spectrophotometric method, using Ellman’s reagent. Reduced glutathione was used as a reference standard. Data were expressed in μg/g tissue (wet weight) and calculated assuming a cysteine content in a mussel metallothionein molecule of 21 residues and a molecular weight of 8600 Da.

Superoxide dismutase (SOD) activity in digestive glands was estimated based on the ability of SOD to inhibit the reduction of oxidized dianisidine by superoxide radical photochemically generated from riboflavin, using a spectrophotometric method [14]. Absorbance was converted to SOD units from an external standard curve of pure bovine erythrocyte SOD. One unit of SOD is defined as the amount of SOD causing 50% inhibition in the reaction between oxidized dianisidine and O_2_^−^.

Reduced glutathione (GSH) concentration was measured with the recycling method in digestive gland deproteinized lysates, using 5′dithio-bis-(2-nitrobenzoic acid), as described by Pan et al. [48]. Total GSH (tGSH) was determined by adding to the reaction mixture nicotinamide adenine dinucleotide phosphate (NADPH) and glutathione reductase. After incubation of the reaction for 20 min, the oxidized glutathione (GSSG) content was estimated by subtracting the amount of GSH from the amount of tGSH. Absorbance for the determination of GSH was read at 412 nm using UV-vis spectrophotometer. GSH and tGSH were quantified using a standard curve of known concentrations of GSH [48]. Data were expressed in nmol per mg of protein.

### 2.3. Biochemical Preparations

A. Total cytosolic proteins

Digestive glands (3g) were homogenized with a Teflon/glass potter homogenizer at 4 °C in three volumes of homogenization buffer containing 20 mM Tris–HCl pH 7.6,150mM ammonium chloride, 10 mM magnesium acetate, 0.5 mM ethylenediaminetetraacetic acid (EDTA), 58 μg/mL phenylmethylsulfonyl fluoride (PMSF), 250 mM sucrose, and 6 mM β-mercaptoethanol. Cell-free lysates were obtained by two sequential centrifugations at 13,000× *g* for 20 min. Total proteins were isolated from the cell-free lysate after trichloroacetic acid/acetone precipitation [49], centrifugation at 13,000× *g* for 20 min (4 °C), and washing with cold acetone. The pellet was solubilized in lysis buffer containing 7 M urea. The Bradford assay was used to determine the concentration of total proteins [50].

B. FWR Protein Fraction

Cell-free lysates were centrifuged at 100,000× *g* for 7.5 h at 4 °C. The pellet including ribosomes and polysomes was treated with puromycin (0.5 mM) in the presence of 0.5 M ammonium chloride [15]. After a centrifugation at 100,000× *g* for 7.5 h at 4 °C, the pellet was used for ribosomal preparation [10], while the supernatant was concentrated with ammonium sulfate treatment and used as a fraction enriched in translation factors, which is named the Washed Ribosomal Factors (FWR) fraction.

C. Ribosomal Proteins

The previous pellet after centrifugation in 100,000× *g*, as mentioned in Section 2.3, was treated to isolate 80S proteins. Briefly, the pellet was dissolved in buffer (50 mM Tris-HCl, pH 7.6, 2 mM Mg(CH_3_COO)_2_, 50 mM KCl, 6 mM β-ETSH) and extraction with acetic acid followed according to Barritault [51]. RNA was removed through ethanol precipitation, and ribosomal proteins were pelleted with acetone.

### 2.4. DNPH Dot Blot Assay for theQuantification of Carbonylated Proteins

Dot blot analysis was used for carbonylated proteins measurement. Protein samples (5 μg protein per slot) untreated or treated with NaBH_4_ (20 mM NaBH_4_ at 37 °C for 30 min, neutralization with 2N HCl, ribosomal and overnight dialysis against Phosphate-Buffered Saline (PBS) at 37 °C) were spotted onto a polyvinylidene difluoride (PVDF) membrane, using the slot blotter Bio-Dot SF provided by Bio-Rad Laboratories. Then, PVDF membranes were washed with PBS buffer, following the manufacturer’s protocol, after which they were reacted with 0.5 mM 2,4-dinitrophenylhydrazine (DNPH; Sigma-Aldrich, St. Louis, MO, USA). Non-specific binding sites were blocked by two incubations in PBS-T (50 mM phosphate buffer pH 7.4, 0.1% (*v/v*) Tween 20) containing 5% (*w/v*) milk powder. After blocking, membranes were incubated overnight at 4 °C with primary antibody, rabbit anti-DNP (1:1000 in PBS-T, Sigma). After washing with PBS-T three times, membranes were incubated with the appropriate secondary antibody (goat anti-rabbit IgG conjugated with horseradish peroxidase, diluted 1:4000; Upstate, Lake Placed, NY, USA). Carbonylated proteins were visualized by incubation with ECL^TM^ chemiluminescence reagent (Amersham) and detection by autoradiography. The intensity of bands was quantified by Image Analysis, using the Image-Pro Plus 7 software (Media Cybernetics, Rockville, MD, USA). The intensities of immunostained bands, corrected by subtraction of the intensity of the corresponding reduced samples, were converted to nmoles of carbonyl groups using a standard curve of bovine serum albumin (BSA) samples differentially oxidized [52]. Results were expressed as carbonyl nmol/mg of protein. It should be noted here that the intensity of a spot depends on the amount of carbonyl groups but also on the time of membranes exposure to the Fuji Medical X-Ray film. Therefore, each assay of a protein sample was accompanied by the assay of BSA standards, as analyzed in parallel.

### 2.5. WesternBlot Detection of Specific Proteins

This method was used to determine the expression levels of translation factors. First, 20 μg of FWR fraction or ribosomal proteins were combined with equal volumes of 2× loading buffer (120 mM Tris-HCl pH 6.8, 4% sodium dodecyl sulfate (SDS), 20% glycerol, 100 mM β-mercaptoethanol (β-ETSH), 0.1% bromophenol blue) and incubated at 90 °C for 5 min. Then, samples were fractionated on 12% SDS-PAGE gels under reducing conditions (70 V for 2 h, 4 °C) and transferred to PVDF blots (Amersham Biosciences) using an iBlot transfer apparatus (Invitrogen). Membranes were stained with 0.2% Ponceau S in 5% acetic acid to check for equal protein loading, transfer, and blotting efficiency. After destaining, membranes were pre-treated for post-electrophoretic detection of carbonylated proteins by washing in 20% methanol/80% PBS-T (50 mM phosphate buffer pH 7.4, 0.1% (*v/v*) Tween20) and equilibrating them in 2N HCl. Then, membranes were incubated with 0.5 mM 2,4-dinitrophenylhydrazine (DNPH; Sigma-Aldrich) for 10 min in the dark. Then, the derivatized membranes were washed with 2N HCl (twice, 10 min per wash). For the detection of translation factors, membranes were blocked and incubated with rabbit polyclonal anti-eIF2α (anti-eukaryotic InitiationFactor 2α) or anti-eEF1A1, (anti-eukaryotic Elongation Factor 1A1) (1:1000 in PBS-T, Aviva Systems Biology Corp. San Diego, CA, USA), or the following antibodies from Cell Signaling Technology, diluted 1/1000: rabbit monoclonal anti-phospho-4E-BP1 (Thr37/46), polyclonal anti-phospho-4E-BP1 (Thr70), polyclonal anti-eIF4E, polyclonal anti-phospho-eIF4E (Ser209), polyclonal anti-phospho-eIF2α (Ser51), polyclonal anti-eEF2, and polyclonal anti-phospho-eEF2 (Thr56). Incubation of the membranes with the secondary antibody and detection of the immunoreactive proteins was performed as described previously.

### 2.6. Statistical Analysis

Apart from electrophoresis runs (three replicates), all other assays were repeated five times, and the data were expressed as the mean ± SD. To examine the differences between groups, one-way ANOVA was applied after testing normality by a one-sample Kolmogorov–Smirnov test. Pairwise comparisons were performed by post-hoc F-Scheffé test, after testing the equality of error variances by Levene’s test. The level of significance was set at *a* = 0.05.

## 3. Results

### 3.1. Stress Biomarkers

#### 3.1.1. Metal Bioaccumulation

The accumulation of the three metals in the digestive glands of mussels is shown in Figure 1. All three metals were bioaccumulated extensively, although with different rates and extents. Cd showed the highest concentration following a saturation curve and reached almost a plateau after 10 days of exposure, while both Cu and Hg showed a slower rate of accumulation and needed more than 15 days to be saturated. Neither mortality nor milder clinical signs were observed during the first 10 days of exposure. Mortality of less than 5% was recorded only on the 14th day of exposure to 100 μg/L Cd^2+^.

#### 3.1.2. Oxidative Damage Indicators

To assess the oxidative status of mussels after exposure to trace metals, we measured four established oxidative damage indicators: micronucleus frequency; superoxide radical production; stability of lysosomal membrane; and lipid peroxidation. The results are summarized in Table 1.

All biomarkers have been importantly modified almost from the fifth day of exposure, establishing a state of oxidative damage. Micronucleus frequency was increased, confirming the genotoxicity caused by Cu and Hg [53]. Cd failed to increase the micronuclei frequency. Superoxide radical production was gently modified by Cu and Hg, but in the case of Cd, it increased by ten times. Labilization periods were reduced in a similar way, while malondialdehyde increased slightly by Cu and Hg and almost three times by Cd.

#### 3.1.3. Anti-OxidantBarrier

We measured superoxide dismoutase activity, metalothionein levels, and glutathione concentration. Superoxide dismutase activity had a similar fluctuation for all metals, but metallothioneins had a completely different pattern. In the case of Cu, the levels of metallothioneins remained constant. In the presence of Hg, they were modified only after the 15th day of exposure, and in the case of Cd, they increased from the 5th day to the 15th day by almost seven times. Cu and Hg depleted the intracellular pools of tGSH and reduced GSH, causing a significant decrease in the GSH/GSSG ratio, which serves as a reliable index of oxidative stress [54]. Cd induced a sharp depletion (50%) of GSH during the first five days; however, prolonged exposure (15 days) led to an elevation of GSH levels in the digestive gland that exceeded the control values. In turn, the elevation of both MTs and GSH content diminished the early oxidative stress, thus stabilizing a new quasi-stationary phase of mild oxidative intensity.

### 3.2. Dot Blot Analysis of Carbonylated Proteins

The method of choice for the measurement of carbonyl groups in our samples was a dot blot immunoassay, as it provides high sensitivity compared to the spectrophotometric assay, without loss of proteins during the preparation of samples and without interferences with nucleic acids [55]. To eliminate the non-specific binding of the anti-DNP antibody on our samples, reduced blanks were prepared by treatment with 20 mM NaBH_4_, and their staining intensities subtracted from those of NaBH_4_-untreated samples [52]. Two representative autoradiograms are shown in were Figure 2, while the results of carbonyl group analysis obtained in several cellular fractions are summarized in Table 2. According to our findings, the level of carbonyl groups in the total protein samples is the highest among the various fractions in control mussels. The FWR fraction exhibited an intermediate carbonylation profile, while proteins isolated from ribosomes constituted the slightest carbonylated population. This was more evident for mussels treated with metals. The exposure of mussels to metals for 5 days increased the carbonyl group content of proteins, with the effect caused by Hg and Cd being more pronounced. However, while the level of carbonyl groups increased at prolonged exposure of mussels to Cu or Hg, the effect of Cd during the late period of exposure (15 days) became milder (Table 2).

### 3.3. WesternBlot Analysis

A.Carbonylated FWR

Although the analysis of carbonyl groups by the dot-blot immunoassay is a quantitative approximation to estimate the oxidative damage in the protein fractions, it does not provide any information on the extend of oxidation of a particular protein in these complex mixtures, nor can it reveal any alteration on the amount of each protein. For this reason, the protein fractions were further analyzed by electrophoresis combined with Western blot immunoassays. Only FWR and ribosomal proteins were further analyzed because their content is more defined and concentrated in contrast to cell-free lysate, which contains a huge variety of proteins. Both fractions were first separated with electrophoresis and then evaluated either with color staining (Coomassie or Ponceau) (Figure 3A and Figure 4A) or Western blotting for DNP (Figure 3B and Figure 4B). Gels stained with Coomassie blue reveal that there are alterations in the expression of some specific bands which correspond to individual proteins, although loaded proteins are equal (Figure 3A). These are signed by an asterisk and denote underlining differences in the expression of specific translation factors when mussels are exposed to metals. FWR carbonylation according to dot-blot immunoassay (Table 2) was increased approximately twice in Cu-15 days samples (Figure 3B), three times in Hg-15 days samples, and two times in Cd-15 days samples compared to the control. The same tend of alterations were also seen (Figure 3), although the method is not a quantitative one. It is important to emphasize again the carbonylation level fluctuation caused by Cd, according to which there was a sharp early increase in the 5th day, but a severe decrease was achieved by the 15th day.

B. Carbonylated Ribosomal Proteins

Figure 4 depicts an analogous analysis for ribosomal proteins either with Ponceau staining (Figure 4A) or Western blotting (Figure 4B). The results are consistent with the findings of the dot-blot analysis. More specifically, exposure to metals increased carbonylation, which is more pronounced in mussels exposed to Hg and Cd for 5 days (Figure 4B). At 15 days of exposure, carbonylation increased in the mussels exposed to Cu and Hg. However, the same trend was not observed in the case of Cd, where carbonylation values showed a decline. An equally interesting observation is that the density of bands has changed (Figure 4A), supporting the suggestion that ribosomal proteins stoichiometry is not one by one in all ribosome population, but in contrast, there is an heterogeneity among the ribosome population, which is very important for gene regulation and organismal life [56].

### 3.4. Translation Factors Expression

Considering that translation factors under stress not only undergo changes in their expression or oxidation status but are also regulated by several modifications, such as phosphorylation, hydroxylation, and acetylation among others, reviewed by [57], our study was expanded using antibodies against a series of translation factors that follow an orchestrating quick adaptation to metal environmental challenges that mussels may face. Representative Western blots are given in Figure 5.

As shown in panel A of Figure 5, the expression of translation factor eIF2α remained constant and similar to that of the control. However, the phosphorylation of eIF2α after 15 days of exposure was increased, which is a modification that leads to the general inhibition of the initiation step [58].

Another translation factor implicated in the initiation phase of protein synthesis and investigated here is eIF4E. eIF4E binds to the 5′ cap structure of mRNAs, thus allowing the eIF4G complex to anchor on the starting tail of the mRNA chain. As shown in Figure 5B, the expression of eIF4E was upregulated by metals, particularly by Cd after 15 days of exposure. On the other hand, p-eIF4E is upregulated after being exposed to Cd for 15 days (Figure 5B; bottom panel). Given that eIF4E-binding proteins (4E-BPs) interfere in the regulation of eIF4E, they were our next target. We found that in control mussels and those exposed to Cu for 5 days, the phosphorylation of 4E-BP1 was plenty (Figure 5C, upper panel). However, exposure to Hg or Cd led to the attenuation of phosphorylation, with a subsequent harmful impact on eIF4E complex assembly and protein synthesis. Phosphorylation after 15 days exposure to Cd was restored to approximately normal levels. Similar alterations were observed with another antibody for the same protein, which is appropriate for phosphorylation on the Ser65 position (Figure 5, bottom panel).

As for eEF1A1, we found that the exposure of mussels to Cu and Hg caused a decrease in eEF1A1 levels at 5 days, which became more pronounced after 15 days (Figure 5D). Cd impact at the 5th day of exposure was similar to those of Cu and Hg, but after 15days exposure to this metal, eEF1A1 levels reverted to control ones, supporting its damage fluctuation.

Concerning Elongation Factor 2, we did not observe significant changes in its expression at protein level, under all conditions studied (Figure 5E). However, we recorded higher levels of phosphorylation of eEF2, particularly after exposure of mussels to Hg for 15 days and to Cd for 5 days. Prolonged exposure of mussels to Cd (15 days) induced a weaker phosphorylation level again in eEF2 (Figure 5F).

## 4. Discussion

According to our data, all three metals bioaccumulate in the mussels’ digestive glands and cause oxidative stress, the severity of which depends on the time of exposure and the nature of each metal. The metal concentrations used were sublethal (lower than IC50) but high enough to cause apparent metal bioaccumulation in the indicated times without extensive mortality [59,60], while similar concentrations have been recorded in highly polluted areas [61,62]. In addition, the same exposure conditions have been reported by several field and laboratory studies [13,26,63,64], but this is the first time correlating metal ions oxidative damage through the glasses of both traditional biomarkers and a protein synthesis machine. The exposure of mussels to cadmium has led to oxidative stress in mussels, which is characterized by an early increase in superoxide radical production, lipid peroxidation, decreased lysosomal membrane stability, and DNA damage. As a result, antioxidant defense mechanisms including SOD and metallothioneins were activated. On the other hand, copper and mercury caused milder oxidative stress according to all biomarkers, except for MN increase, confirming their serious genotoxic effect (Table 1). Proteins are involved in key functions of the cell and provide important structural and functional advantages in several cellular organelles, such as ribosomes. The accumulation of oxidatively damaged proteins is a hallmark of deleterious processes. Thus, quantifying and identifying oxidized proteins provide fundamental insights into the underlying mechanism of toxicity. Translation factors and ribosomal proteins catalytically contribute to ribosomal function and are among the most potent compounds for inflicting oxidative damage on cells [33]. Compared to other oxidative protein products, the formation of carbonyl groups is the most recognized biomarker of oxidative damage in proteins, which is mainly due to the product chemical stability [55]. The level of carbonylation depends on the source of proteins (Table 2). The total proteins lysate showed higher amounts of carbonylation compared to the FWR. Ribosomal proteins were the least carbonylated fraction. This suggests that ribosomal proteins in the statement of the assembled ribosomes are protected from the toxic action of ROS, while translation factors bound on the ribosomes surface but not assembled exhibited medium carbonylation exposure. In the case of exposed mussels, Cu and Hg showed their maximum carbonylation level at 15 days of exposure, while Cd showed its most toxic effect in 5 days, which was reversed at the end of the exposure period (Table 2). In general, mussels can reverse the initial oxidative damage by Cd inducing antioxidant defense mechanisms, such as the production of MTs and elevation of GSH levels. The next part of the study focused on the expression and integrity of translation factors and ribosomal proteins, which are actively involved in the formation and function of the ribosome. It becomes clear that the accumulation of lesions can directly affect protein synthetic capacity. The electrophoretic run of FWR and ribosomal protein fractions revealed that apart from protein carbonylation, protein expression is also modified under metal exposure (Figure 3A) and additionally, the heterogeneity of ribosomes is a real and new important finding under these conditions. Moreover, the carbonylation profile of the various protein fractions is depending also on the idiosyncratic features of each metal.

Protein synthesis is regulated in eukaryotic organisms by many translation factors, which are subject to strict regulation by signal transduction pathways, responding to environmental stress. Regarding the factors involved in the initiation step, eIF2α and eIF4E were studied in parallel with their phosphorylation levels. eIF2 is a component of the ternary complex Met-tRNAi-GTP-eIF2 that is associated with the 40S ribosomal subunit to form the 43S preinitiation complex (PIC). eIF2 is regulated by phosphorylation on the eIF2α subunit. All eIF2α kinases become activated in response to oxidative stress [65], and this explains why p-eIF2α is upregulated at the late stages of the mussel response (Figure 5). eIF4E is the most abundant and important protein of all the initiation factors, and its overexpression under oxidative stress has been associated with the selective synthesis of antioxidant proteins, which prevent the abnormal accumulation of ROS [66]. eIF4E is regulated via phosphorylation and through eIF4E-binding proteins (4E-BPs), which interfere with eIF4F assembly by impairing eIF4E-eIF4G interaction [67]. There are three 4E-BP isoforms (4E-BP1, 2, and 3), with 4E-BP1 being more abundant than 4E-BP2 and 4E-BP3 [68]. In response to environmental stress, 4E-BPs are phosphorylated in multiple residues and in a hierarchical fashion [69]. Additionally, two translation factors participating in the elongation of protein synthesis were studied. eEF1A1, as a complex with GTP, delivers aminoacyl-tRNAs into an empty A-site of an elongating ribosome. The complementarity of codon–anticodon interaction stimulates GTP hydrolysis and releases eEF1A1–GDP from the ribosome. We found that a long exposure of mussels to metals caused a decrease in eEF1A1 levels, except for the exposure to Cd, where eEF1A1 levels reverted to the control ones. This agrees with previous studies that found a downregulation of the eEF1A1 gene in yeast [51], fish *Pinephalespromelas* [70], and *M. galloprovincialis* [30], which was challenged by metals. Peptide bond formation leaves a deacylated tRNA in the P-site and a newly formed peptidyl-tRNA in the A-site. This triggers eEF2–GTP binding and the translocation of ribosomal substrates from the A- and P- sites to the P- and E-sites, respectively. The upregulation and downregulation of eEF2 at the transcription level have been frequently found in plants and animals, including mussels [31,71]. However, gene expression alone is not informative enough to define the effects of metals on proteins, due to post-transcriptional, translational, and post-translational regulation. The phosphorylation of eEF2 at Thr56 reduces its affinity for GTP and decreases its capability of engaging the ribosome to push translocation on [72]. The high phosphorylation level found in our study (Figure 5F) is indicative of protein synthesis perturbation and is compatible with other oxidative stress markers.

## 5. Conclusions

Trace metals are serious stressors in aquatic ecosystems that can interfere with various physiological processes in the marine organism mussel *Mytilus galloprovincialis*. In a previous study, we evaluated the impact of these metals on the RNA modifications of the protein synthesis machinery [10]. In this project, we explored the damage in the same machinery under the same exposure conditions from the viewpoint of protein expression and modification. Examining different subcellular fractions, we found that proteins are dangerously modified and equally important; they are also upregulated at the translation level. Therefore, trace metals evoke oxidative damage through various stimulations, leading to translation reprogramming. Our data offer innovative perspectives in the field of redox balance under metal exposure through the proteomics approach and can be used as a potential biomarker indicative of the toxicity level. Additionally, the changes in protein expression pattern together with protein modification can serve as a great tool for elucidating the mechanisms of toxicity and could be integrated in biomonitoring programs. Our next goal will be the use of advanced state-of-the-art technology, utilizing high-resolution liquid chromatography coupled to tandem mass spectrometry (LC-MS/MS) to perform a global protein expression analysis of mussels digestive glands grown in the absence and presence of trace metals. These precise data will give us the chance to develop a new intelligent algorithm to correlate mussel protein damage with seawater pollution.

## Figures and Tables

**Figure 1 toxics-08-00089-f001:**
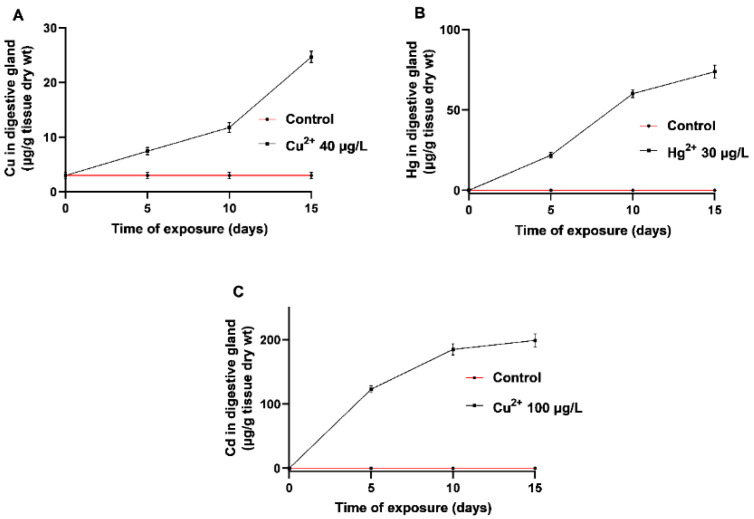
Metal concentrations in digestive glands after mussels exposed for up to 15 days to seawater enriched with different concentrations of (**A**) CuCl_2_, (**B**) HgCl_2_, and (**C**) CdCl_2_. After acclimation at 18 °C for 1 week in tanks containing natural seawater from a non-polluted area, mussels were exposed to natural seawater or to seawater containing 40 μg/L Cu, 30 μg/L Hg, or 100 μg/L Cd. Depicted concentration values represent the means from five independently performed experiments, while standard deviations were lower than 5%.

**Figure 2 toxics-08-00089-f002:**
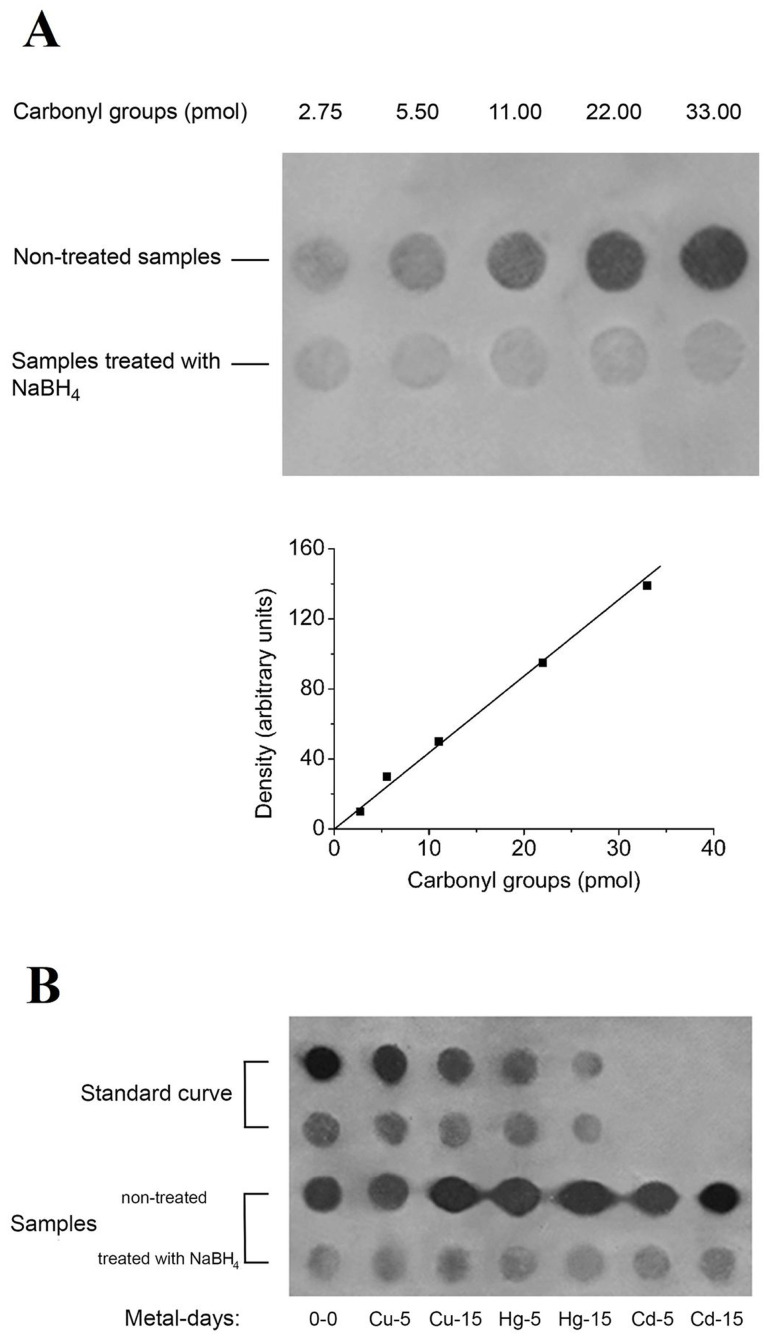
(**A**) A representative standard curve of 2,4-dinitrophenylhydrazine (DNPH) dot blot analysis used for the quantification of carbonylated proteins. The curve was constructed using a series of BSA samples differentially oxidized. The intensities of immunostained bands were corrected by subtraction of the intensity of the corresponding reduced samples. The content of BSA standards in carbonyl groups was estimated by a calorimetric assay [52]. (**B**) Representative autoradiograms of carbonyl group analysis using a dot blot immunoassay for ribosomal proteins of 80S.

**Figure 3 toxics-08-00089-f003:**
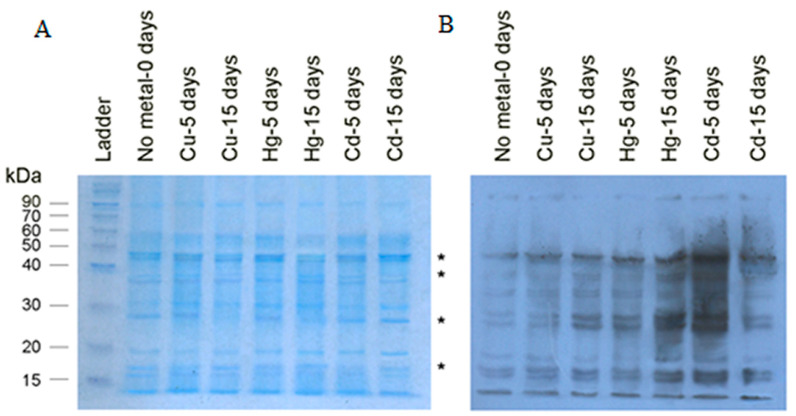
Profile of carbonylated proteins in the fraction of translation factors (Washed Ribosomal Factors, FWR) investigated by Coomassie staining (**A**) and Western blotting for DNP (**B**) in 12% SDS-PAGE gels isolated from non-exposed (no metal-0 days) or exposed mussels to 40 μg/L Cu, 30 μg/L Hg, or 100 μg/L Cd for 5 or 15 days. * Lanes differed in the intensity of specific bands.

**Figure 4 toxics-08-00089-f004:**
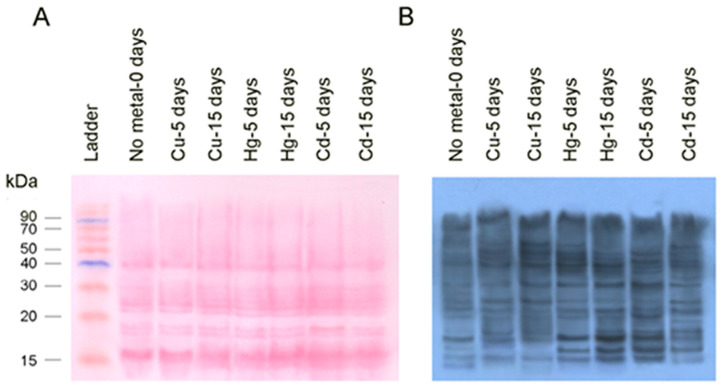
Electrophoretic run of ribosomal proteins and immunoblotting using anti-DNPH as primary antibody: (**A**) Ponceau S staining; (**B**) profile of carbonylated proteins in the fraction of ribosomal proteins isolated from non-exposed (no metal, 0 days) or exposed mussels to 40 μg/L Cu, 30 μg/L Hg, or 100 μg/L Cd for 5 or 15 days.

**Figure 5 toxics-08-00089-f005:**
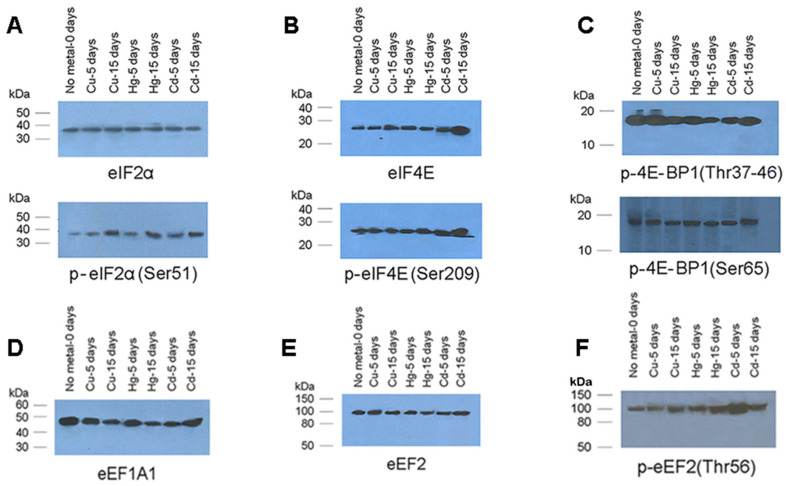
Western blots demonstrating translation factors’ abundance. (**A**) eIF2αand p-eIF2α (Serine51), (**B**) eIF4E and p-eIF4E (Serine209), (**C**) p-4E-BP1 (Threonine37/46) and p-4E-BP1 (Serine65), (**D**) eEF1A1, (**E**) eEF2, (**F**) p-eEF2(Thr56)of exposed mussels to 40 μg/L Cu, 30 μg/L Hg, or 100 μg/L Cd for 5 or 15 days (pstands for the phosphorylated translation factor, and the aminoacid mentioned in parentheses indicates the site of phosphorylation).

**Table 1 toxics-08-00089-t001:** Biomarker measurements in digestive gland of mussels non-exposed or exposed to 40 μg/L Cu, 30 μg/L Hg, or 100 μg/L Cd for 5 or 15 days ^a^.

		Time of Exposure (Days)		
Parameter	Metal	0	5	15
*Oxidative damage*				
MN frequency (ppt *)	Cu Hg Cd	2.6 ± 0.5 2.8 ± 0.8 2.6 ± 0.5	5.6 ± 0.5 ^b^ 6.0 ± 0.7 ^b^ 3.4 ± 0.9	10.0 ± 1.0 ^b^ 12.0 ±1.6 ^b^ 4.0 ± 1.2 ^b^
SR (pmol/mg protein)	Cu Hg Cd	2.6 ± 0.2 2.5 ± 0.1 2.6 ± 0.2	1.5 ± 0.2 ^b^ 2.7 ± 0.1 ^b^ 8.0 ± 1.0 ^b^	0.4 ± 0.1 ^b^ 2.6 ± 0.2 22.4 ± 2.1 ^b^
LP (min)	Cu Hg Cd	28.0 ± 2.0 30.0 ± 3.2 27.0 ± 1.9	16.0 ± 1.6 ^b^ 18.0 ± 1.6 ^b^ 9.0 ± 1.4 ^b^	9.0 ± 1.6 ^b^ 11.0 ± 1.0 ^b^ 6.5 ± 0.5 ^b^
MDA (nmol/mg protein)	Cu Hg Cd	2.0 ± 0.3 2.1 ± 0.3 2.0 ± 0.2	2.2 ± 0.2 2.4 ± 0.1 2.6 ± 0.2 ^b^	2.8 ± 0.3 ^b^ 2.6 ± 0.2 ^b^ 6.2 ± 0.4 ^b^
*Anti-oxidant barrier*				
SOD (units/mg protein)	Cu Hg Cd	0.8 ± 0.1 0.7 ± 0.1 0.8 ± 0.1	0.9 ± 0.1 1.8 ± 0.2 ^b^ 1.6 ± 0.2 ^b^	0.2 ± 0.1 ^b^ 0.6 ± 0.1 0.6 ± 0.1 ^b^
MTs (μg/g tissue w.w)	Cu Hg Cd	52.0 ± 5.0 50.0 ± 4.5 50.0 ± 4.5	56.0 ± 5.3 50.0 ± 4.5 210.0 ± 21.6 ^b^	53.0 ± 3.9 85.0 ± 7.4 ^b^ 354.0 ± 30.5 ^b^
GSH (nmol/g tissue w.w.)	Cu Hg Cd	740.0 ± 67.3 780.0 ± 60.3 720.0 ± 69.9	511.0 ± 44.6 ^b^ 565.0 ± 60.0 ^b^ 369.0 ± 40.1 ^b^	421.0 ± 45.3 ^b^ 275.0 ± 38.1 ^b^ 852.0 ± 98.2 ^b^
GSSG (nmol/g tissue w.w.)	Cu Hg Cd	117.0 ± 16.9 122.0 ± 13.7 112.0 ± 14.4	104.5 ± 10.2 110.5 ± 10.3 76.0 ± 6.9 ^b^	102.0 ± 10.0 95.5 ± 8.0 ^b^ 23.0 ± 2.9 ^b^
GSH/GSSG ratio	Cu Hg Cd	6.32 ± 1.1 6.39 ± 0.9 6.42 ± 1.0	4.89 ± 0.6 ^b^ 5.11 ± 0.7 4.85 ± 0.7 ^b^	4.13 ± 0.6 ^b^ 2.88 ± 0.5 ^b^ 37.04 ± 6.3 ^b^

MN, micronucleus; SR, superoxide radical; LP, labilizationperiod; MDA, malondialdehyde; SOD, superoxide dismutase; MTs, metallothioneins; GSH, reduced glutathione; GSSG, oxidized glutathione. ^a^ Each value is expressed as mean ± S.D. (*n* = 5). ^b^ Significantly different value from that measured in non-exposed mussels. * Micronucleated cells per 1000 cells.

**Table 2 toxics-08-00089-t002:** Levels of carbonyl groups (nmol/mg protein) in the proteins of a digestive gland in non-exposed and exposed mussels to 40 μg/L Cu, 30 μg/L Hg, or 100 μg/L Cd for 5 or 15 days.

		Time of Exposure (Days)		
Cellular Component	Metal	0	5	15
Total cytosolic proteins	Cu Hg Cd	7.8 ± 1.2 7.5 ± 0.8 7.2 ± 0.8	22.0 ± 2.4 ^a^ 44.7 ± 4.3 ^a^ 52.6 ± 6.6 ^a^	48.5 ± 4.2 ^a^ 53.6 ± 6.2 ^a^ 28.4 ± 4.1 ^a^
Proteins FWR fraction	Cu Hg Cd	7.5 ± 0.6 6.8 ± 0.6 6.9 ± 0.7	10.5 ± 1.5 ^a^ 13.9 ± 1.4 ^a^ 18.4 ± 1.7 ^a^	17.3 ± 2.2 ^a^ 18.6 ± 2.4 ^a^ 12.2 ± 2.3 ^a^
Proteins in 80S ribosome	Cu Hg Cd	2.8 ± 0.2 3.1 ± 0.3 2.8 ± 0.3	3.4 ± 0.3 ^a^ 5.8 ± 0.4 ^a^ 6.2 ± 0.6 ^a^	7.4 ± 0.7 ^a^ 8.1± 0.8 ^a^ 4.0 ± 0.4 ^a^

^a^ Significantly different values from those recorded in non-exposed mussels (*p* < 0.05).

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
