# Peer review of "Oxidative Damage of Mussels Living in Seawater Enriched with Trace Metals, from the Viewpoint of Proteins Expression and Modification"

_toxics, 2020, doi:10.3390/toxics8040089_

Round 1

Reviewer 1 Report

The authors follow up their previous findings on the effect of trace metals, such as Cu, Hg and Cd, on mussels living in seawater. While the previous study focussed on the effects of RNA, here the authors focus on protein synthesis. There is growing evidence that oxidative damage through various stimuli evokes various stress responses in eukaryotes leading to translational reprogramming. The authors show here that trace metals also evoke a similar response and also provide a series of outputs that can be used for biomonitoring.

Overall, the results are well-conducted and presented, the conclusions are well-supported by the data. There are however a number of minor grammatical corrections that would improve the readability of the manuscript if corrected. Perhaps its worth getting a native speaker to read through the manuscript once, however, many could be corrected by the authors for example, missing spaces on line 122, 146, 354, 68, 369

Also genus and species should be in italics? e.g. line 404 and 416.

Lastly, there is some inconsistency with the use of eEF1A or eEF1a?

Author Response

We thank Reviewer 1 for his/her positive comments and advice for our MS improvement. We tried to answer all points step by step.

Point 1: Perhaps its worth getting a native speaker to read through the manuscript once, however, many could be corrected by the authors for example, missing spaces on line 122, 146, 354, 368, 369

Response 1: We got help from an English native speaker and all points were done.

Point 2: Also genus and species should be in italics? e.g. line 404 and 416.

Response 2: Were done

Point 3: Lastly, there is some inconsistency with the use of eEF1A or eEF1a?

Response 3: Its our fault, we have now replaced eEF1a with correct eEF1A1.

Reviewer 2 Report

General remarks
The biomarkers should be grouped as heavy metals, antioxidant barrier, oxidative damage. Why were only protein oxidation products evaluated?

Introduction
Some fragments are a repetition of the previous ones, read the text carefully. Underline the importance of research.

Material and methods
Specify exactly how many specimens took part in the experiment.
How were the doses of heavy metals determined? How was the incubation time established? Describe in more detail the conditions of the experiment.
2.4. Give the principle of each method.
2.7. Justify the use of the F-Scheffe test. How was the normal distribution checked?

Results
Figure 1: The data in the line graphs should be presented as mean and standard deviation.

Discussion
The discussion does not end with clear conclusions. What is new in this research?
What are the limitations of the study? What is the next step?

Author Response

Response to Reviewer 2 Comments

We thank Reviewer 2 for his/her careful and constructive review of our manuscript. Overall, we have followed this referee's suggestions positive comments and advice for our MS improvement. We tried to answer all points step by step.

Point 1: General remarks

The biomarkers should be grouped as heavy metals, antioxidant barrier, oxidative damage. Why were only protein oxidation products evaluated?

Response 1: In parallel with protein oxidation products we have also evaluated classical biomarkers like reactive oxygen species (ROS), glutathione levels, metallothionein expression etc. Our interest is the comparison of usual biomarkers with proteins expression and/or modification and not the evaluation of protein oxidation products.

Point 2: Introduction

Some fragments are a repetition of the previous ones, read the text carefully. Underline the importance of research.

Response 2: We read again the introduction carefully but did not find any repetition

Material and methods

Point 3a: Specify exactly how many specimens took part in the experiment.

Response 3a: Τwenty specimen took part. We added the number in line 97 (new line numbering)

Point 3b: How were the doses of heavy metals determined? How was the incubation time established?

Response 3b: The concentration of trace metals exposure was selected enough high in order to reach the appropriate level after short time of treatment and the incubation time was also kept short to maintain high ions concentrations without mortality, according to previous studies using differential doses of trace metals (Pytharopoulou et al., 2011). This paragraph is added in lines 93-95 (new numbering).

Point 4:

Describe in more detail the conditions of the experiment.

2.4. Give the principle of each method

Response 4: All methods are now described in principle, giving the appropriate references. New sentences and /or paragraphs are added as following:

  1. Lysosomal membrane stability was calculated by measuring the time in minutes required to destabilize the lysosomal membranes, under acid conditions [45]. This protocol is a cytochemical procedure and is based on the evaluation of the activity of N-acetyl-β-hexosaminidase, a lysosomal enzyme. Lines 145-147.
  2. Micronucleus (MN) frequency was determined in gill cells, according to Bolognesi and Fenech [46]. The relative occurrence of micronuclei can provide an indication of accumulated genetic damage in marine organisms even during short phases of contamination. Lines 149-150.
  • Metallothionein (MT) content in digestive gland lysates was measured, following the protocol and assumptions made by Viarengo et al. [47]. The method is briefly based on a metallothionein fractionation and SH residue content quantification by a spectrophotometric method, using Ellman’s reagent. Lines 152-154.
  1. Lipid peroxidation in digestive gland lysates was estimated by determining the thiobarbituric reactive substances [48], based on the reaction with the chromogenic reagent, N-methyl-2- phenylindole, with malondialdehyde and 4-hydroxyalkenals. Lines 156-157.
  2. Superoxide radical (·O2-) production in vivo was calculated using an ultrasensitive fluorescent assay for the in vivo quantification of superoxide radical according to Georgiou et al. [50] and were expressed in pmoles produced per mg tissue protein in 30 min. Lines 161-162.
  3. Superoxide dismutase (SOD) activity in digestive gland was estimated based on the ability of SOD to inhibit the reduction of oxidized dianisidine by superoxide radical, photochemically generated from riboflavin [14]. Lines 164-166.
  • Reduced glutathione (GSH) concentration was measured with the recycling method. Line 168.

Point 5: Justify the use of the F-Scheffe test. How was the normal distribution checked?

Response 5: The use of F-Scheffe test was justified with the check of equal variances through the Levene’s test and the normality was checked with Kolmogorov-Smirnov test. We added the relevant phrase in 2.7 section. Lines 218-221.

Results

Point 6:  Figure 1: The data in the line graphs should be presented as mean and standard deviation.

Response 6: The figure is replaced with a new one showing the standard deviation

Discussion

Point 7: The discussion does not end with clear conclusions. What is new in this research?

Response 7: Conclusions have been modified and a new paragraph has been added.

Therefore, trace metals evoke oxidative damage through various stimulation leading to translation reprogramming. Lines 440-441.

Point 8: What are the limitations of the study? What is the next step?

Response 8: Our limitation is the missing from the literature of a complete mussel protein bank to correlate exactly data from advanced proteomics study with trace metal levels.

Our next-goal will be the use of advanced state-of-the-art technology, utilizing high resolution liquid chromatography coupled to tandem mass spectrometry (LC-MS/MS) to perform global protein expression analysis of mussels digestive glands grown in the absence and presence of trace metals. These precise data will give us the chance to develop a new intelligent algorithm to correlate mussel protein damage with seawater pollution.

This paragraph is added in the text lines 445-450.

Round 2

Reviewer 2 Report

The authors did not respond exactly to all the comments.

The biomarkers should be grouped as heavy metals, antioxidant barrier, oxidative damage (both in methods and results).

Verses 93-95: give citations.

2.4 Give the principle of each method. Specify exactly whether the colorimetric/fluorimetric method was used, what was assumed as one unit of enzyme activity, etc.

Verse 145-147: Give the correct citation.

Author Response

We thank Reviewer for his/her positive comments and advice for our MS improvement. We tried to answer all points step by step.

Point 1: The biomarkers should be grouped as heavy metals, antioxidant barrier, oxidative damage (both in methods and results).

Response 1: We have now grouped oxidative biomarkers according reviewer’s suggestion. Both in methods and results have been presented as: Heavy metals, antioxidant barrier (metallothionein, superoxide dismutase and glutathione), and oxidative damage (Lysosomal membrane stability, micronucleus frequency, lipid peroxidation and superoxide radical).   New lines are added with new highlight color.

Lines  72-75, 121, 145, 237, 252, 263.

Point 2: Verses 93-95: give citations.

Response 2: The appropriate reference has been added, number 14 and in line 93.

Point 3: 2.4 Give the principle of each method. Specify exactly whether the colorimetric/fluorimetric method was used, what was assumed as one unit of enzyme activity, etc.

Response 3: Each method has been described in details, and new insertions are highlighted. Lines: 122, 125-127, 129-131, 136-137, 141-142, 149-151, 154-155.

Point 4:   Verse 145-147: Give the correct citation.

Response 4: The correct reference  is added number 45 and lines